# Healthcare providers' (HCPs) perspectives in initiating discussion on mammogram screening, and their perceived barriers and enablers to screening in women—A qualitative study

Xuefei Wang[1]*, Qi He Mabel Leow[1], Hwee Khim Lee[1], Ching Yee Mok[1], Wei Teen Wong[1], Hui Min Joanne Quah[1,2], Ngiap Chuan Tan[1,2]

1 SingHealth Polyclinics, Singapore, Singapore, 2 SingHealth- Duke NUS Family Medicine Academic Clinical Programme, Singapore, Singapore

* wang.xue.fei@singhealth.com.sg

## Abstract

### Background

Breast cancer is the most common cancer in women, and mammogram screening can reduce breast cancer mortality. Healthcare providers' perspectives can have an impact on encouraging females to attend mammogram screening.

### Objective

To understand healthcare providers' (HCPs) perspectives in initiating discussion on mammogram screening, and their perceived barriers and enablers to screening in women.

### Method

A qualitative descriptive study design was used. Focus group discussions (FGD) were conducted with a purposive sample of doctors and nurses from SingHealth Polyclinics between February 2019 to July 2020. "The Generalists' Wheel of Knowledge, Understanding and Inquiry" was adopted as the conceptual framework to design the interview guide and analyse the data.

### Results

Thirty participants consisting of 14 doctors and 16 nurses were interviewed across five FGDs. HCPs personal factors such as gender, their personal experience with mammogram, and years of work experience influenced counselling. They perceived that patients' language, education status, previous experience with mammogram, misconceptions, and fear of mammogram results were potential barriers to mammogram. HCPs believed they had the knowledge to provide information and initiate discussions on mammogram screening.

**Data Availability Statement:** All relevant data are within the manuscript and its Supporting Information files.

**Funding:** This study was funded by the SingHealth Polyclinics Research Support Program – SEED funding (Grant No. SHP-SEED49-2018) received by Wang Xuefei.

**Competing interests:** The authors have declared that no competing interests exist.

Healthcare factors were cost, accessibility to screening, consultation time constraints, and inadequate information in health records.

## Conclusion

HCPs' information mastery and personal experiences were enablers in encouraging females to attend mammogram screening. Lack of information in the health records was a barrier to providing continuity of care.

## Implications for practice

Nurses should be opportunistic in initiating conversations and encourage women on mammogram screening when they visit the clinic. Training and being updated on mammography screening guidelines is important for clinicians to initiate and provide discussions on mammogram screening.

## Introduction

Breast cancer is the most common cancer with 2.3 million women diagnosed with breast cancer in 2020, and ranks 5[th] in mortality with 685,000 deaths worldwide [1]. In Singapore, breast cancer also led with an incidence of 29.4% and mortality of 17.1% between 2015 and 2019 [2]. Mammography is currently the gold standard for breast cancer screening. A Cochrane systematic review [3] reported that mammogram screening reduces breast cancer mortality by 15%, with other studies reporting a reduction in mortality by 20–40% [4,5].

The screening rates for breast cancer are 77% and 61% respectively in the United States (US) and the United Kingdom. In Singapore, breast cancer screening can be performed at both public and private tertiary hospitals or general practice clinics [6,7]. All public general practice clinics, also known as Polyclinics, have a multidisciplinary team of healthcare workers, and provide a wide range of services such as phlebotomy, radiology, rehabilitation, nutrition, and podiatry services. The majority of general practice private clinics are family-owned, and typically only provide medical consultation, dispensing of medications, and simple nursing procedures. They can refer patients to other private clinics for additional services. Patients who visit both public and private clinics require to pay a fee, but the fee at the public clinic is subsidised by the government. In addition, there are also voluntary welfare organisations that provide free screening. In Singapore, A National report [8] found that only 38.7% of eligible women attended screening in 2019, which is lower than that reported in other Western developed countries. This low rate is in spite of a nationwide breast screen program that provides free or subsidised rates for screening [9], and encourages women to attend mammogram screening [10]. The National report [8] found that women who received tertiary education (50%) were more likely to attend screening compared to those with primary education (28%). Women in Singapore expressed that the barriers to mammogram screening included 1) fear of the breast cancer screening procedure and the sequel of having a positive result, 2) personal challenges such as having no time, inconvenience of screening location, and paying for screening and treatment, 3) modesty and embarrassment, and 4) low perceived susceptibility to breast cancer [11].

Healthcare providers (HCPs) play a role in providing information and initiating discussions with females to attend mammogram screening. However, HCP might face challenges

that hinder their ability to provide such discussion. Four main challenges encountered by physicians were identified—1) time constraints 2) lack of adequate knowledge of risks and ability to communicate risk in an effective format 3) confusion on the latest guidelines and 4) personal preferences in addressing patient preferences that contradict guidelines and addressing physician's own biases) [12,13]. It is also not known if the individual practices of HCPs would influence their approach to initiate topics on mammogram screening. In Singapore, the National Health Promotion Board provides education to the public on mammogram screening through their website. HCPs might also provide information to women opportunistically when they visit the doctor. A survey of nurses working in public health in Singapore revealed that the nurses had high knowledge on breast cancer and screening, but it did not affect their practice on screening as only 50% went for screening [14]. Similar trend was found in a study conducted in Nigeria [15].

From a review of the literature, most studies [11,16] understand the barriers to mammogram screening from the women's perspectives. It is important to understand the barriers that HCP face as they play an important role in initiating discussions and guiding women's decision-making process [12]. This study aims to understand HCPs perspectives in initiating discussion and encouraging uptake of mammogram screening, and their perceived barriers and enablers to patients' attending mammogram. The findings of this study will be used to improve current strategies in increasing uptake of mammogram screening, and training HCPs to initiate conversations on mammogram screening.

## Methods

### Study design

A qualitative descriptive study design was used to explore HCPs' views and concerns in initiating discussions and encouraging the uptake of screening mammogram. The interviews were conducted in English using focus group discussions (FGD) between February 2019 to July 2020. An interview guide with broad questions was used to guide the discussion (See S1 File).

### Study site and participants

The participants were recruited from SingHealth Polyclinics (SHP), which consists of a network of ten subsidised public primary care facilities located in the Eastern region of the island state. It is also an accredited academic training center for Family Medicine in Singapore. The polyclinics in total see about 10,000 patients a day.

A purposive sampling was used to recruit doctors and nurses of various ages and years of experience to obtain various perspectives. The inclusion criteria included a healthcare provider who: (1) was an employee of the polyclinic, (2) was a doctor or a nurse, (3) was age 21 and above and (4) was able to provide informed consent. Recruitment was performed until data saturation was met, which meant that no new information was obtained from the sample [17].

### Theoretical framework

"The Generalists' Wheel of Knowledge, Understanding and Inquiry" was adopted as the conceptual framework to design the interview guide and analyse the data [18]. The framework encompasses four factors "clinician", "patient", "disease" and "system", and describes the relationships between these domains (Fig 1). In the primary care setting, all 4 factors ("clinician", "patient", "disease" and "system") play a significant role in the uptake of the mammogram. Through HCP's experiences interacting with patients and the local healthcare system, they can provide their views and concerns under the "clinician" domain. "Integration" which is at the

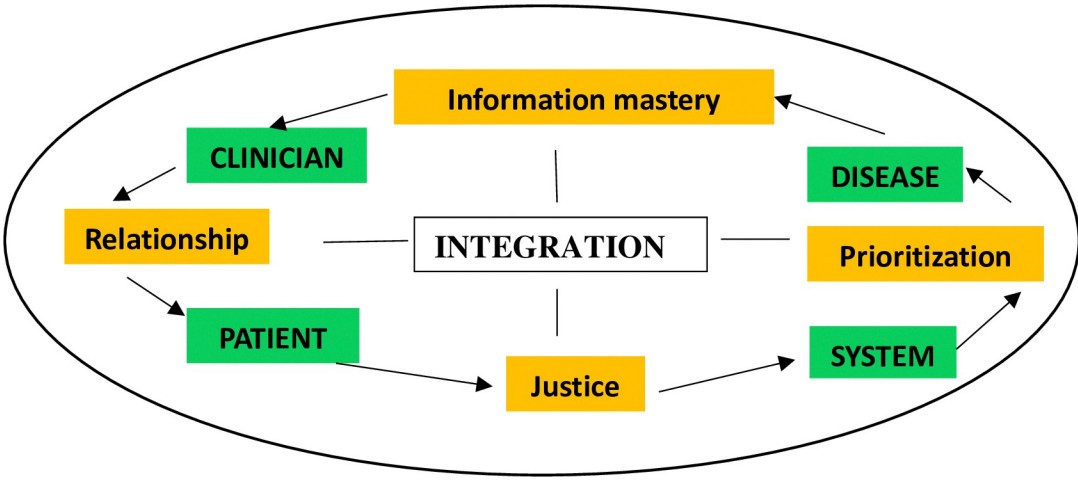

**Fig 1. The Generalists' Wheel of Knowledge, Understanding and Inquiry.**

center of the framework allows a succinct and relational presentation of the themes across the domains. This framework has been used in other studies conducted in primary care to explore the experience of primary care physicians in managing postpartum care and vaccine hesitancy in childhood [19–21].

## Data collection

The participants were recruited from February 2019 July 2020 after approval was given by the Centralised Institutional Review Board (CIRB). Participants were provided an information sheet with details on the study, and written informed consent was obtained prior to the interviews.

Participants' socio-demographic characteristics obtained were their profession, age, gender, and years of experience. All FGDs were conducted in a quiet room at the study site, audio-recorded, and lasted between 30 to 50 minutes. Each participant was assigned a study identification number and transcripts were de-identified. WXF and TNC were the main moderators for the FGDs, and the LHK or MQY assisted to take field notes.

## Data analysis and rigour

Transcripts were coded, categorised, and themed using the framework method, following the seven steps proposed by Gale and colleagues [22]. An example of the themalised process is in Table 1. The following seven steps were taken.

Step 1: Transcription

The audio-recording was transcribed verbatim, with cross-checking by the principal investigator to ensure accuracy.

Step 2: Familiarisation with the interview

The team members who were performing the data analysis listened to the audio recording at least two times to familiarize themselves with the interview.

Step 3: Coding

After familiarization, the transcripts were coded independently by three team members (WXF, LHK & MCY). Open coding was first performed to ensure that all important aspects of the data were not missed.

Step 4: Developing a working analytical framework

**Table 1. Example of themalised process.**

| HealthCare Provider | Code | Category | | Theme |
|---|---|---|---|---|
| HCP 1 | Not wanting to know the unknown | Fear of mammogram results | | Patient factors |
| HCP 3 | Their reason (of not going) is that they feel that there are no symptoms | Women's misconceptions | | |
| HCP 3 | I think if some patients who are not well-educated, especially the elderly aunties or the grandmothers, they might not even be able to understand the concept of the false positive, false negative, so the message cannot come across." | Education | | |
| HCP 6 | Pain is a common discouragement factor | Previous experience with mammogram | | |
| HCP 9 | Bad experience when they are having this mammogram | Previous experience with mammogram | | |
| HCP 12 | Not knowing is bliss, so they are fearful | Fear of mammogram results | | |
| HCP 21 | Language barrier is one of it | Language | | |
| HCP 24 | They misunderstand that it's the mammogram that activate(s) the cancer | Women's misconceptions | | |
| HCP 29 | They had it done before and they don't like the procedural discomfort | Previous experience with mammogram | | |

After coding the first few transcripts, all researchers met to compare the labels they have applied and agree on a set of codes to apply to all subsequent transcripts. Codes were grouped together into categories based on the framework which was pre-defined.

Step 5: Applying the analytical framework

Subsequent transcripts were indexed using the existing categories and codes, and the working analytical framework was applied.

Step 6: Charting data into the framework matrix

The data from each transcript were summarized by category to reduce the amount of data while retaining the original meanings of the interviewees' words (See S2 File).

Step 7: Interpreting the data

Characteristics of and differences between the data are identified, and connections between categories to explore relationships and/or causality were mapped. Reasons for the emergence of a phenomenon were explained.

## Rigour

Rigour was established by ensuring credibility, transferability, dependability, and confirmability of data [23]. To ensure credibility in data analysis, triangulation was performed through feedback on the accuracy of the interpretation and analysis from team members after the analysis of each interview. Critical self-reflection was used in data collection and analysis to reduce bias from self-imposed viewpoints [24]. For transferability, thick description of data was reported to provide context and meaning to the text to allow data to be transferred [25]. Dependability was ensured by developing an audit trail, which consisted of raw data, audio recordings, products of data analysis and synthesis, and interview guides, to increase

transparency of the research process [24]. Confirmability was achieved when the criteria of credibility, transferability and dependability were established [23].

## Results

### Sample

Forty-four clinicians were invited to participate in this study. Thirty participated (14 doctors and 16 nurses), representing a take-up rate of 68.2%. Five FGDs were conducted with a group size of two to eleven participants. Doctors and nurses were interviewed separately except one doctor joined the nurse group as she could not make it for doctor's FGDs. 83.3% of the participants were females, aged from 29 to 69 years. Their work experience ranged from 4 years to 50 years. The demographic profile is in Table 2. The findings were grouped into four main themes based on the conceptual framework– 1. Healthcare professional personal factors; 2. Patient factors; 3. Healthcare professional information mastery on mammogram and 4. Healthcare system factors. The areas and subthemes are presented in Fig 2.

**Table 2. Participant demographics.**

| S/N | Profession | Age range | Gender | Years of experience |
|---|---|---|---|---|
| 1 | Doctor | 41–50 | M | 20 |
| 2 | Doctor | 31–40 | F | 15 |
| 3 | Doctor | 31–40 | F | 8 |
| 4 | Doctor | 31–40 | F | 6 |
| 5 | Doctor | 31–40 | F | 5 |
| 6 | Doctor | 31–40 | F | 12 |
| 7 | Doctor | 41–50 | F | 14 |
| 8 | Nurse | 31–40 | F | 10 |
| 9 | Nurse | 31–40 | F | 10 |
| 10 | Nurse | 21–30 | F | 11 |
| 11 | Nurse | 51–60 | F | 15 |
| 12 | Nurse | >60 | F | 46 |
| 13 | Nurse | 31–40 | F | 23 |
| 14 | Nurse | 51–60 | F | 30 |
| 15 | Nurse | 31–40 | F | 10 |
| 16 | Nurse | >60 | F | 43 |
| 17 | Nurse | 31–40 | F | 16 |
| 18 | Nurse | >60 | F | 50 |
| 19 | Nurse | 41–50 | F | 20 |
| 20 | Nurse | 31–40 | F | 10 |
| 21 | Nurse | 51–60 | F | 32 |
| 22 | Nurse | >60 | F | 46 |
| 23 | Doctor | 41–50 | M | 23 |
| 24 | Doctor | 31–40 | F | 7 |
| 25 | Doctor | 31–40 | M | 4 |
| 26 | Doctor | 21–30 | F | 5 |
| 27 | Doctor | 21–30 | F | 4 |
| 28 | Doctor | 21–30 | M | 5 |
| 29 | Doctor | 21–30 | M | 5 |
| 30 | Doctor | 21–30 | F | 3 |

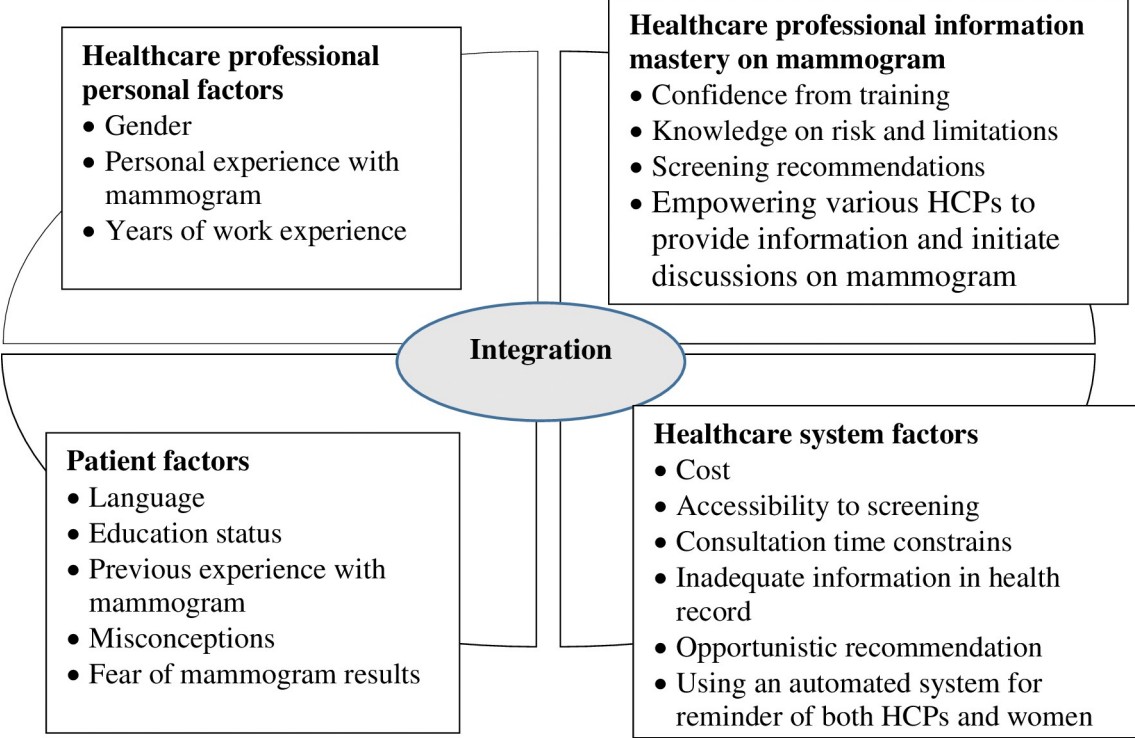

**Fig 2. Factors influencing HCPs' perspectives in initiating discussion on mammogram screening.**

### Healthcare professional personal factors

This theme explores the HCP's perception of mammogram screening. The subthemes are gender, their personal experience with mammogram, and years of work experience.

**Gender.** Male HCPs expressed their gender being a limitation. Although they felt comfortable initiating discussions with women on mammography screening, they acknowledged their limitation of not being able to share any personal experiences. Also, they were aware that some women might prefer having a female doctor to discuss mammography screening with them.

*"Not being female. If I tell a diabetes patient about insulin needles, I can tell the patient I've poked myself with a needle before. But I've not gone through a mammogram before, so I just know that it's very painful, but you know, you can't really (explain further)." (HCP 1)*

**Personal experience with mammogram.** Female HCPs who had gone for mammograms or had testimonies of family members or friends who benefited from screening, felt more confident with initiating discussions on mammogram screening. They could share their personal experience which they believed provided a personal touch and convinced patients to attend screening. Female HCPs who have not gone through mammography screening, felt that they were not able to share much personal experience with other women.

*"I mean, sometimes if I've a little bit more time, I'll share personal experiences, that 'I've gone for mammogram. It's not as painful as you think it is.', or 'I have got a relative who has just got diagnosed recently with very early-stage breast cancer, and it's because it's picked up from*

*early screening mammogram.', so I encourage them on early screening, early detection, early treatment. This kind of thing. . . rather than just telling them from a healthcare provider('s) point of view" (HCP 2)*

**Years of work experience.** HCPs who had less work experience or in having discussions with patients on mammograms expressed their lack of confidence in this area.

*"Actually, I don't feel very confident at the moment. Even though I know this. Because I don't see that many women with. . . I haven't spoken to many women about mammograms. I think it's probably less than 10. Less than one patient a month. . ." (HCP 30)*

**Patient factors.** Patient factors that impacted the mammogram screening were language, education status, previous experience with mammograms, misconceptions, and fear of mammogram results.

**Language.** English is the default lingua franca in Singapore although a wide range of languages, such as Chinese, Malay, and Tamil, can be used. Not all women are proficient in English. Some of them were only able to speak their ethnic mother tongue or dialects which the HCPs were unable to speak. The HCPs felt that the language barrier affected their ability to initiate discussions on mammogram screening.

*"I agree about the language barrier, because majority are Chinese speaking, sometimes it's very hard for us, for those speaking other language, it's very hard to counsel in their language. Yes. That's the major issue, one of the major issue(s)." (HCP 19)*

**Patient's education status.** In addition to the language barrier, the HCPs were concerned that some women might not be able to understand certain information that was provided to them due to their educational level, although the HCPs tried to use lay terms.

*"It actually depends on the patients' age and education level sometimes. In fact, I think if some patients who are not well-educated, especially the elderly aunties or the grandmothers, they might not even be able to understand the concept of the false positive, false negative, so the message cannot come across." (HCP 3)*

**Women previous experience with mammogram.** Many HCPs mentioned that previous mammography screening experience influenced women's decision to repeat screening. The women chose not to repeat if they had unpleasant experiences or negative feelings previously. Pain was the most common bad experience or concern patients expressed to HCPs, which they believed was the reason some women stopped going for mammography.

*"We also have many feedbacks from patient(s) about the pain. That would be like a prior negative experience. . . If they have a bad experience or painful previous screening mammogram, they may not want to do it again. . .Sometimes, some of them actually refuse to go for further screening." (HCP 3)*

**Women's misconceptions.** Women's misconceptions influenced their decision on mammography screening. Some of the reasons included the belief that they would not get breast cancer, being young and healthy, having no breast symptoms, mammogram only needs to be performed once in a lifetime, and being single. As mammogram uses X-ray beams, some women were afraid that performing mammograms would increase their radiation exposure.

*"Like... they believe that breast cancer won't happen to them, so there's no point screening. They self-perceive risk is low." (HCP 30)*

*"Some patients went for it once, and were told it's normal, then subsequently they thought no need to do anymore." (HCP 5)*

**Fear of mammogram results.**   HPCs highlighted that some women declined mammograms as they were fearful of knowing that they had breast cancer from the mammogram.

*"There are some patients who really do not want to know more or find out more; they are fearful of the consequences after the mammogram." (HCP 6)*

**Healthcare professional information mastery on mammogram.**   HCPs were confident in providing information as they had undergone training. It was also suggested to empower various HCPs to provide counselling on mammogram.

**Confidence from training.**   Most of the interviewed HCPs felt confident to provide information and initiate discussions on mammogram screening as they had been adequately trained. They also believed that this was part of their work as an HCP. HCPs were updated on local guidelines and recommendations for mammogram screening and knew where to get updates and information.

*"As a nurse, as a midwife, ... I think I am confident to advise the patient, because these are part of our work as a medical staff to make them aware about mammogram and to show them the proper way of doing breast self-examination and give them advice about this matter." (HCP 12)*

*"because when we are in an institution and when there are changes to guidelines, usually we do get updates, either from the workgroups or the heads." (HCP 3)*

**Knowledge on risks and limitations.**   The majority of HCPs were aware of the risks and limitations of mammograms. They mentioned the possibility of getting false negative and false positive results, as well as the need to discuss this limitation with women. Women with abnormal mammogram findings were required to undergo further investigations.

*"There can be false positives and false negatives, especially in younger age groups, like less than fifty years old. So, for forty to fifty, other than telling them that they need to start breast screening, they need to know that having a mammogram can have a higher chance of having false negatives in this age group, because the breast is more dense. So, they might end up having to go through further investigations, because of abnormal findings, then they must be able to accept that. And then, the investigations can turn out to be normal. So, the investigations can be invasive sometimes. . .And if they are really keen, they should screen for every year." (HCP 7)*

HCPs mentioned the risk of exposure to radiation while doing mammography screening. However, they would still advise women to go for a mammogram as they knew it uses a very low dose of radiation and is less likely associated with negative health effects.

*"The radiation-wise, I think it's not very much. It's very minimal. It's a very short process, so the exposure is very short." (HCP 16)*

**Screening recommendation.** HCPs acknowledged that the screening mammogram is recommended mainly for those women who are at general risk of breast cancer. It might not be the best or most appropriate screening tool for those who are at high risk or special groups of women. HCPs would refer them to a specialist.

*"if she's younger than fifty (years old), then the use of mammography compared to ultrasound is (that) there is a grey area whereby which one picks up abnormal breast lesion better. So, it's still up to the clinician's discretion in terms of which modality suits the patient." (HCP 26)*

HCPs tried to facilitate mammograms by making it convenient for women to attend the screening. They provided the hotline to make an appointment for the screening or encouraged them to get a same-date appointment with other tests.

*"I would have recommended and given the telephone number and so on,. . . I would bundle it together with the PAP smear, with the CVD screening." (HCP 1)*

*". . . encourage them to get the same-date appointment with the PAP smear, so they don't have to come down twice." (HCP 9)*

**Empowering various HCPs to provide information and initiate discussions on mammogram.** The physicians suggested empowering more HCPs to provide information and initiate discussions on mammograms, instead of solely relying on doctors.

*"I think it doesn't have to come from the doctors. Any healthcare providers can provide this information, and they can also help to counsel." (HCP 2)*

**Healthcare system factors.** Barriers to screening that were systemic included cost, accessibility to screening, consultation time constraints, and inadequate information in health records. HCPs expressed that screening could be recommended during clinic visits for an acute condition to increase take-up. An automated system to remind both HCPs and women when their screenings are due.

**Cost.** Cost was cited as a significant contributor to the low take up rate of mammography screening.

*"Probably the cost of screening mammogram. If the cost of the screening mammogram can be lower or even be free for those who are eligible, probably more patients will be willing to go." (HCP 26)*

*". . .however, for those that are really very poor, like living from hand to mouth, I think cost might be a problem. They might put it off in favour of say, spending their money on other things." (HCP 29)*

**Accessibility to screening.** Most of the polyclinics provide mammography screening services. However, the polyclinics are mostly only open during working hours—from Monday to Friday in the daytime, with only half a day in the morning on Saturday. Thus, women who work might be required to take leave to attend mammograms. The waiting time and location could also pose a barrier to screening. HCPs mentioned that the breast screening take-up rate can be increased if the screening can be brought to women instead of waiting for them to come.

*". . .If let's say they conduct a roadshow, with the mobile screening van that can do on the spot, then they can just pick off people from the roadshow and then just screen them directly, I think that will be quite convenient for the patient. And can increase the breast screening rates also, roadshow with screen on the spot." (HCP 25)*

**Opportunistic recommendation for mammogram.** HCPs suggested that it is important to be opportunistic in encouraging women to go for mammograms, such as when they attend the polyclinic for other acute or chronic problems. However, they also acknowledged that this topic is less likely to be raised as it is not the patient's main concern for the visit.

*"Because most of the time for the patients, the screening mammograms is an opportunistic kind of thing; most of the time they do come to clinic for other problems, chronic or acute problems." (HCP 3)*

**Consultation time constraints.** One common challenge that physicians faced was the time constraints. In clinical practice, about 5 to 10 minutes is allocated for each consultation. Physicians would prioritise the patient's main reason for the visit and did not have time to address preventive care due to the short consultation time.

*"I mean, consult time. . . very short consult time. So, it is probably not possible to advise every single one of eligible female for breast cancer because of the (short) clinic consult." (HCP 23)*

**Inadequate information in health records.** The current electronic health records (EHRs) are not linked Nationally, thus HCPs were not able to check if the women had performed the mammogram if it was done in another healthcare organisation. This affected their ability to provide care continuity and follow-up on the women's mammogram status. Most of the time, HCPs had to rely on the women to provide their mammography screening status, but not every woman kept a record of their last screening.

*". . . it's only visible if you're done in SingHealth. It's on the EMR (Electronic Medical Record). If it's done in another hospital, or in NHG (National Healthcare Group), or private, you wouldn't be able to see it at all." (HCP 30)*

*"most of them would say that they'll go and do it. But whether they actually do it or not, we wouldn't really know, because our systems are not linked, as mentioned earlier." (HCP 27)*

**Using an automated system to remind both HCPs and women when screening is due.** HCPs felt it would be helpful if they could receive reminders to inform the women to attend mammograms. In addition, they found it useful when the health promotion board was involved in sending out reminders to women to attend mammogram when it is due.

*"Prompting. if you have the IT (system) prompting, like, this patient is due for mammogram, it helps, to remind the doctor as well in a busy clinic." (HCP 23)*

*"I think the Health Promotion Board does send out this screening, so I think that is also a good way to let them know that they can go for it and where (to go for screening)" (HCP 28)*

## Discussion

This qualitative study aimed to understand HCPs' perspectives on initiating discussions on mammogram screening, and their perceived barriers and enablers to attending mammogram. Using the framework analysis method, the findings were categorised into four main areas– 1. Healthcare professional personal factors; 2. Patient factors; 3. Healthcare professional information mastery on mammogram; and 4. Healthcare system factors. A framework analysis was used to identify the factors specific to each area, so that interventions can be developed to address the issues.

In a previous literature review conducted in Singapore [14], the patient factors and healthcare system factors highlighted were misconceptions, fear of mammograms results, cost, and accessibility to screening. However, the issues with accessibility and cost had been addressed using mammogram buses managed by voluntary welfare organisation which went to the neighbourhoods to perform screening free of charge. In this study, additional patient factors such as language barrier, education status, and previous negative experiences with screening were identified. Although English is regarded de facto as the main language in Singapore, many women of older age, or foreign migrants, might not be able to speak English, or be proficient in the language. In such situations, counselling for mammogram would ideally be provided by an HCP who is able to speak the women's native language.

The HCPs expressed that women who were older and had lower education were not able to understand the information provided to them on mammograms. This finding concurred with a survey study of 3,245 women in Saudi Arabia [26]. However, women with higher education and income might face challenges in terms of remembering appointments (53%) and time to get a mammogram (24%) [27]. Thus, challenges to attending mammograms faced by women with low and high income are different but are in existence. Hence, different strategies are required to address the needs of women from various demographics, such as one-on-one or group education, assistance in scheduling screening, sending reminders, and engaging the help of community health workers [28,29].

Barriers to mammogram has largely been explored through the women's lens. However, it is important to note that HCP factors can have an impact on counselling for mammograms. In this study, HCPs who had had mammogram and had testimonies of family members or friends gaining positive benefits mammogram, felt that they were able to share their personal experiences which would make their counselling more persuasive. On the contrary, those who had not done mammogram felt that they could only provide the limited factual information they were taught. Thus, HCPs' own practice may influence patients' choice for mammogram screening. Further, HCPs who had longer work experience and counselled more patients were more likely to be confident in providing counselling. Martinez and colleagues [30] found that HCPs with more than five years of work experience reported higher confidence in providing counselling for a mammogram.

In this study, the HCPs generally agreed that they were well-trained to provide counselling on mammogram, were aware of the risks and benefits of screening, and were kept up to date with the screening recommendations. They also knew where they could go to look for information if required. In our institution, HCPs have access to an intranet with all healthcare-related information. There are also workgroups who regularly update the latest guidelines, and share information with all staff in the institution. In a study in the United States [12], physicians in the private general practice expressed a lack of knowledge on the risks of mammogram and confusion over most recent guidelines. In Singapore, it is not known how the HCPs in private general practice settings keep up with the mammogram guidelines. It might be important to grant them access to some of this information available to public healthcare institutions.

Doctors and nurses are the HCPs who most commonly provide information and initiate discussions on mammogram, playing a pivotal role in encouraging screening [31]. However, this is not always possible due to time constrains in the clinical setting. Thus, it was suggested that all HCPs can be involved in encouraging women to attend mammogram. Considering time constraints, offering pamphlets could be the quickest way to provide information. A previous local study found that the distribution of prints from healthcare organisations and providers, as patients rated them the preferred, and most highly trusted sources of breast cancer information [32]. These pamphlets could provide the seed to women initiating conversations on mammogram with their HCP, or attending a mammogram screening in the near future. Ancillary staff in the healthcare institution can also be trained to provide information to women on mammogram screening, and facilitate further discussions if required.

Lack of information in health records was cited as a barrier to providing continuity of care, of which mammogram screening is one of them. In Singapore, mammograms can be performed in several primary and tertiary institutions both in public and private, and voluntary welfare organisations. This poses a challenge to allow all health information to be deposited in a single health record system to provide alerts when the screening was due, and for HCP to track easily. Thus, the responsibility to attend mammogram regularly will still largely rest on the women. As such, it is important to educate women on the importance of regular mammogram screening.

## Strengths and limitations

To our knowledge, it is one of the first studies to focus on HCPs' perspectives in initiating discussion on mammogram screening in Singapore and conducted in the primary care setting. The primary care HCPs have the first–touch advantage in counselling for cancer screening. This study included doctors and nurses, which brought in the perspectives of the various HCPs.

We recognise the limitations of our study. Although primary care is provided by public and private practice, the HCPs in private practice were not included, which could limit our findings.

## Clinical implications

Women need to be provided information and educated on the importance of attending mammograms regularly based on guidelines. Due to time constraints in the clinical setting, an information tool such as a decision aid can be developed to support women in their decision to attend mammograms. Ancillary staff can also be trained to provide information and initiate discussion on mammogram for women with general risk for breast cancer. Dedicated consultation time would be needed to ensure that there is ample time for HCPs to discuss and address patients' misconceptions and fears.

Visiting a regular primary care practice could aid in the follow-up on preventive care including mammogram screening and results. However, not all women have a regular primary care practice. Thus, it is important for nurses to be opportunistic in initiating discussions on mammogram screening when they visit the clinic.

At the national level, having a fully subsidised national screening program, and screenings brought to the women's vicinity can help overcome barriers due to cost and convenience. If possible, a Nationalised electronic medical record with information on patients' mammogram status and an alert reminder when the screening is due, can help HCPs remind patients to attend the screening.

## Conclusion

This study generated insights on HCPs' perspectives to initiate discussions on mammogram screening. HCPs' information mastery and personal experiences with mammogram screening were enablers in initiating these discussions. Lack of information in the health records was a barrier to providing continuity of care. Mammogram counselling should be provided in the women's native language, with brochures being provided in various languages at the health-care setting. A buddy system can be initiated, whereby HCPs who are new in providing mammogram counselling can do so with another experienced colleague to build confidence. Ancillary staff in the healthcare institution can also be trained to provide information to women on mammogram screening, and facilitate further discussions if required.

## Supporting information

**S1 File. Interview guide.** This is an interview guide with broad questions was used to guide the discussion.
(PDF)

**S2 File. FGDs transcript.** The data from each transcript were summarized by category based on the pre-defined framework.
(XLSX)

**S3 File. Participant demographics data.** The demographic data from thirty participants.
(XLSX)

## Acknowledgments

We thank all the doctors and nurses who have generously contributed their views and perspectives in our interviews. We would also like to thank Caris Tan and the team from the Department of Research in SingHealth Polyclinics for their assistance throughout the study.

## Author Contributions

**Conceptualization:** Xuefei Wang, Wei Teen Wong, Hui Min Joanne Quah, Ngiap Chuan Tan.

**Formal analysis:** Xuefei Wang, Hwee Khim Lee, Ching Yee Mok, Ngiap Chuan Tan.

**Investigation:** Xuefei Wang, Hwee Khim Lee, Ching Yee Mok, Ngiap Chuan Tan.

**Methodology:** Xuefei Wang, Ngiap Chuan Tan.

**Supervision:** Xuefei Wang, Ngiap Chuan Tan.

**Validation:** Xuefei Wang, Hwee Khim Lee, Ching Yee Mok.

**Writing – original draft:** Xuefei Wang.

**Writing – review & editing:** Xuefei Wang, Qi He Mabel Leow, Hwee Khim Lee, Ching Yee Mok, Wei Teen Wong, Hui Min Joanne Quah, Ngiap Chuan Tan.

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
