## [Decision Letter · Decision Letter 0]

22 Aug 2024

PONE-D-23-32350Healthcare providers’ (HCPs) perspectives in initiating discussion on mammogram screening, and their perceived barriers and enablers to screening in women – a qualitative studyPLOS ONE

Dear Dr. Wang,

Thank you for submitting your manuscript to PLOS ONE. After careful consideration, we feel that it has merit but does not fully meet PLOS ONE’s publication criteria as it currently stands. Therefore, we invite you to submit a revised version of the manuscript that addresses the points raised during the review process.

We look forward to receiving your revised manuscript.

Kind regards,

Maleika Heenaye- Mamode Khan

Academic Editor

PLOS ONE

“

•            WXF

•            S$5000

•            SingHealth Polyclinics Research Support Program – SEED funding (SHP-SEED49-2018)

•            https://polyclinic.singhealth.com.sg

•            No

“

3. We note that your Data Availability Statement is currently as follows: [All relevant data available from PI.]

Reviewers' comments:

Reviewer's Responses to Questions

**Comments to the Author**

1. Is the manuscript technically sound, and do the data support the conclusions?

Reviewer #1: Partly

Reviewer #2: Yes

2. Has the statistical analysis been performed appropriately and rigorously? 

Reviewer #1: N/A

Reviewer #2: N/A

3. Have the authors made all data underlying the findings in their manuscript fully available?

Reviewer #1: Yes

Reviewer #2: Yes

4. Is the manuscript presented in an intelligible fashion and written in standard English?

Reviewer #1: Yes

Reviewer #2: Yes

5. Review Comments to the Author

Reviewer #1: • Paragraphing to justify in abstract

• In the Introduction “Breast cancer is the commonest cancer with 2.3 million women diagnosed with breast cancer in 2020, and ranks 5th in mortality with 685,000 deaths”. Need to mention these info. are for the whole world.

• The literature has been well surveyed.

• The research methodology used has been described quite well with detail discussion about the no. of participants, the type of interviews, the themes and coding structure.

• Figure 1 and Figure 2 are not appendixes. They should be placed under their respective section at 2.3 and 2.4.

• Paragraphing to justify in the last part of the paper before the reference section.

• However, the conclusion section is quite poor. One of the aim of the study is to encourage uptake of mammogram screening. Recommendations of how to proceed with encouraging women to go for the screening have not been detailed. Similarly how to initiate the discussion has not been well discussed.

• Overall the paper is well structured, but the authors need to provide more guidance on how to initiate discussion and encourage women to go for mammography and how to overcome the barriers faced.

Reviewer #2: This study employed qualitative methods to explore healthcare professionals' views on the barriers and facilitators to mammogram breast screening uptake in Singapore. Doctors and nurses participated in focus groups. The findings highlighted several factors influencing screening uptake, such as language barriers, women's educational backgrounds, and cost/accessibility.

While the study addresses a significant topic, several improvements are needed:

Typos: The document contains several typos that require correction.

Introduction: Provide contextual information about Singapore's healthcare system, as PLOS ONE's global audience may be unfamiliar with it.

Literature Update: Data collection stopped in 2020. The literature review should be updated to include more recent studies.

Methods: Further explanation of the theoretical framework is necessary. How widely used is this framework, and why was it chosen over others?

Results: Detail the number of participants in each of the five focus groups. Indicate whether there was a mix of doctors and nurses, as group dynamics could influence the findings.

Discussion: The strengths and limitations section should include some strengths.

6. PLOS authors have the option to publish the peer review history of their article (what does this mean?). If published, this will include your full peer review and any attached files.

Reviewer #1: No

Reviewer #2: **Yes: **Louisa Lawrie

---

## [Author Response · Author response to Decision Letter 0]

11 Sep 2024

PLOS ONE review

Reviewer #1: 

• Paragraphing to justify in abstract 

Response: Agree. We have, accordingly, justified the abstract. 

• In the Introduction “Breast cancer is the commonest cancer with 2.3 million women diagnosed with breast cancer in 2020, and ranks 5th in mortality with 685,000 deaths”. Need to mention these info. are for the whole world. 

Response: Agree. We have amended it accordingly.

Breast cancer is the commonest cancer with 2.3 million women diagnosed with breast cancer in 2020, and ranks 5th in mortality with 685,000 deaths worldwide.1

• The literature has been well surveyed.

Response: Thank you 

• The research methodology used has been described quite well with detail discussion about the no. of participants, the type of interviews, the themes and coding structure. 

Response: Thank you

• Figure 1 and Figure 2 are not appendixes. They should be placed under their respective section at 2.3 and 2.4.

Response: Thank you for pointing this out. We have amended Figure 1 to 2.3, Figure 2 to 3.1 respectively. 

• Paragraphing to justify in the last part of the paper before the reference section.

Response: Agree. We have, accordingly, justified the part.

• However, the conclusion section is quite poor. One of the aim of the study is to encourage uptake of mammogram screening. Recommendations of how to proceed with encouraging women to go for the screening have not been detailed. Similarly how to initiate the discussion has not been well discussed.

Response: Thank you for pointing this out. We agree with this comment. Therefore, we have added the following into the conclusion.

Mammogram counselling should be provided in the women’s native language, with brochures being provided in various languages at the healthcare setting. A buddy system can be initiated, whereby HCPs who are new in providing mammogram counselling can do so with another experienced colleague to build confidence. Ancillary staff in the healthcare institution can also be trained to provide information to women on mammogram screening, and facilitate further discussions if required.

• Overall the paper is well structured, but the authors need to provide more guidance on how to initiate discussion and encourage women to go for mammography and how to overcome the barriers faced.”

Response: Thank you for pointing this out. We have added the following into Discussion paragraph 3.

Hence, different strategies are required to address the needs of women from various demographics, such as one-on-one or group education, assistance in scheduling screening, sending reminders and engaging the help of community health workers.28,29 

28, Nduka IJ, Ejie IL,Okafor CE, et al. Interventions to increase mammography screening uptake among women living in low- income and middle- income countries: a systematic review. BMJ Open 2023;13:e066928. doi:10.1136/ bmjopen-2022-066928

29, Lu M, Moritz S, Lorenzetti D, et al. A systematic review of interventions to increase breast and cervical cancer screening uptake among Asian women. Lu et al. BMC Public Health 2012, 12:413 http://www.biomedcentral.com/1471-2458/12/413

Reviewer #2:

This study employed qualitative methods to explore healthcare professionals' views on the barriers and facilitators to mammogram breast screening uptake in Singapore. Doctors and nurses participated in focus groups. The findings highlighted several factors influencing screening uptake, such as language barriers, women's educational backgrounds, and cost/accessibility.

While the study addresses a significant topic, several improvements are needed:

Typos: The document contains several typos that require correction. 

Response: Thank you. We have amended.

Introduction: Provide contextual information about Singapore's healthcare system, as PLOS ONE's global audience may be unfamiliar with it. 

Response: Thank you for pointing this out. We agree with this and have added in Introduction paragraph 2:

In Singapore, breast cancer screening can be performed at both public and private tertiary hospitals or general practice clinics. All public general practice clinics, also known as Polyclinics, has a multidisciplinary team of healthcare workers, and provide a wide range of services such as phlebotomy, radiology, rehabilitation, nutrition, and podiatry services. Majority of general practice private clinics family-owned, and typically only provide medical consultation, dispensing of medications, and simple nursing procedures. They can refer patients to other private clinics for additional service. Patients who visit both public and private clinics require to pay a fee, but the fee at the public clinic is subsidised by the government. In addition, there are also voluntary welfare organisations which provides free screening.

We have added the following in methods 2.2 Study site and participants:

The participants were recruited from SingHealth Polyclinics (SHP), which consists of a network of ten subsidised public primary care facilities located in the Eastern region of the island state. It is also an accredited academic training center for Family Medicine in Singapore. The polyclinics in total see about 10,000 patients a day. 

Literature Update: Data collection stopped in 2020. The literature review should be updated to include more recent studies.

Response: Thank you. We agree and have added the following

15. Raji MO, Adamu SP, Innibosun-Raji HO, et al. Knowledge, attitude and uptake of mammography among female health workers in two tertiary health facilities of Sokoto state, Nigeria. International Journal of Community Medicine and Public Health. Int J Community Med Public Health. 2021 Feb;8(2):511-517 

16. Tavakoli B, Feizi A, Alavijeh FZ & Shahnazi H. Factors influencing breast cancer screening practices among women worldwide: a systematic review of observational and qualitative studies. BMC Women’s Health (2024) 24:268 https://doi.org/10.1186/s12905-024-03096-x

Methods: Further explanation of the theoretical framework is necessary. How widely used is this framework, and why was it chosen over others? 

Response: Thank you for pointing this out. We have added the following into 2.3 Theoretical framework.

In the primary care setting, all 4 factors (“clinician”, “patient”, “disease” and “system”) play a significant role in the uptake of mammogram. Through HCP’s experiences interacting with patients and local healthcare system, they can provide their views and concerns under the “clinician” domain. “Integration” which is at the center of the framework allows a succinct and relational presentation of the themes across the domains. This framework has used in other studies conducted in primary care to explore the experience of primary care physicians in managing postpartum care and vaccine hesitancy in childhood. 19-21

19. Poon Z, Lee ECW, Ang LP, Tan NC. Experiences of primary care physicians managing postpartum care: a qualitative research study. BMC Fam Pract. 2021 Jun 30;22(1):139. doi: 10.1186/s12875-021-01494-w.

20. Hasnan S, Tan NC. Multi-domain narrative review of vaccine hesitancy in childhood. Vaccine. 2021 Apr 1;39(14):1910-20.

21. Lee AEJ, Ithinin S, Tan NC (2024) Physician factors affecting patient preferences in selecting a primary care provider: A qualitative research study in Singapore. PLoS ONE 19(3): e0298823. https://doi.org/10.1371/journal.pone.0298823

Results: Detail the number of participants in each of the five focus groups. Indicate whether there was a mix of doctors and nurses, as group dynamics could influence the findings.- 

Response: Thank you for pointing this out. We have added the following into 3.1 Sample.

Five FGDs were conducted with a group size of two to eleven participants. Doctors and nurses were interviewed separately except one doctor joined the nurse group as she could not make it for doctor’s FGDs.

Discussion: The strengths and limitations section should include some strengths.

Response: Thank you! We have added the following 

To our knowledge, it is one of the first studies to focus on HCPs’ perspectives in initiating discussion on mammogram screening in Singapore and conducted in the primary care setting. The primary care HCPs have the first – touch advantage in counselling for cancer screening. This study included both doctors and nurses, which brought in the perspectives of the various HCPs.

---

## [Editor Report · Decision Letter 1]

29 Sep 2024

Healthcare providers’ (HCPs) perspectives in initiating discussion on mammogram screening, and their perceived barriers and enablers to screening in women – a qualitative study

PONE-D-23-32350R1

Dear Dr. Wang,

We’re pleased to inform you that your manuscript has been judged scientifically suitable for publication and will be formally accepted for publication once it meets all outstanding technical requirements.

Kind regards,

Maleika Heenaye- Mamode Khan

Academic Editor

PLOS ONE
---

## [Editor Report · Acceptance letter]

11 Nov 2024

PONE-D-23-32350R1 

PLOS ONE

Dear Dr. Wang, 

I'm pleased to inform you that your manuscript has been deemed suitable for publication in PLOS ONE. Congratulations! Your manuscript is now being handed over to our production team.

Kind regards, 

on behalf of

Dr. Maleika Heenaye- Mamode Khan 

Academic Editor

PLOS ONE